# Frontal Encephalocele Plus Epilepsy: A Case Report and Review of the Literature

**DOI:** 10.3390/brainsci13010115

**Published:** 2023-01-09

**Authors:** Ken Yamazaki, Kohei Kanaya, Takehiro Uda, Tetsuhiro Fukuyama, Makoto Nishioka, Yumi Hoshino, Tomoki Kaneko, Ridzky Firmansyah Hardian, Daisuke Yamazaki, Haruki Kuwabara, Kohei Funato, Tetsuyoshi Horiuchi

**Affiliations:** 1Department of Neurosurgery, Shinshu University School of Medicine, Matsumoto 390-8621, Japan; 2Division of Epilepsy, Shinshu University Hospital, Matsumoto 390-8621, Japan; 3Department of Neurosurgery, Osaka Metropolitan University Graduate School of Medicine, Osaka 545-8585, Japan; 4Department of Pediatrics, Shinshu University School of Medicine, Matsumoto 390-8621, Japan; 5Department of Medicine (Neurology and Rheumatology), Shinshu University School of Medicine, Matsumoto 390-8621, Japan; 6Department of Radiology, Shinshu University School of Medicine, Matsumoto 390-8621, Japan

**Keywords:** encephalocele, refractory epilepsy, intraoperative electroencephalography

## Abstract

An encephalocele is a pathological brain herniation caused by osseous dural defects. Encephaloceles are known to be regions of epileptogenic foci. We describe the case of a 44-year-old woman with refractory epilepsy associated with a frontal skull base encephalocele. Epilepsy surgery for encephalocele resection was performed; however, the epilepsy was refractory. A second epilepsy surgery for frontal lobectomy using intraoperative electroencephalography was required to achieve adequate seizure control. Previous reports have shown that only encephalocele resection can result in good seizure control, and refractory epilepsy due to frontal lobe encephalocele has rarely been reported. To the best of our knowledge, this is the first report of frontal encephalocele plus epilepsy in which good seizure control using only encephalocele resection was difficult to achieve. Herein, we describe the possible mechanisms of encephalocele plus epilepsy and the surgical strategy for refractory epilepsy with encephalocele, including a literature review.

## 1. Introduction

Encephaloceles are pathological herniations of the brain parenchyma caused by osteo-dural defects of the skull base or cranial vault, which are acquired or congenital [1]. Encephalocele is diagnosed using ultrasonography, computed tomography (CT), and magnetic resonance imaging (MRI) with MR venography and MR angiography [2]. Encephaloceles cause seizure disorders and medically refractory focal epilepsy [3]. Surgical treatment is generally essential to cure drug-resistant epilepsy [4]. Refractory epilepsy due to frontal lobe encephalocele is rare compared to refractory epilepsy due to temporal lobe encephalocele [5]. Generally, these are treated with resection surgery [6]. To the best of our knowledge, all previous reports on refractory epilepsy caused by frontal lobe encephalocele have been cured by encephalocele resection. However, in our case, resection of the frontal lobe encephalocele did not cure the patient’s refractory epilepsy and required frontal lobectomy. Here, we report a case of encephalocele plus epilepsy.

## 2. Case Presentation

A 44-year-old woman with a history of intellectual disability presented to our institution with medically refractory epilepsy associated with a frontal encephalocele. She had no prior history of head trauma. Seizures began at 41 years of age and were characterized by bilateral tonic-clonic seizures. Antiseizure medication (ASM) (valproic acid 400 mg/day) was started thereafter. Atonic seizures and atypical absence seizures appeared at 42 years of age and progressed intractably, although two ASMs were used (levetiracetam 2000 mg/day and valproic acid 600 mg/day). She had a history of cerebrospinal fluid (CSF) rhinorrhea; however, there were no obvious clinical symptoms suggesting meningitis. MRI showed a 3 cm right frontal encephalocele with a prolapsed right rectal gyrus through the anterior skull base defect (Figure 1A,B). Therefore, resection of the encephalocele and skull base reconstruction were performed as a treatment for CSF rhinorrhea and refractory epilepsy (Figure 1C). Pathological examination of the encephalocele showed inflammatory cell infiltration, including foam cell infiltration. However, daily atypical absence seizures persisted despite the use of multiple ASMs after the first surgery. Ictal video electroencephalogram (EEG) showed an epileptogenic focus on the right frontal region. 18F-fluorodeoxyglucoce PET showed hypometabolism in right lateral frontal cortex (Figure 1D). A second surgery was scheduled to treat intractable epilepsy. A right frontal lobectomy was performed using an intraoperative electrocorticogram (ECoG). ECoG was recorded with a bandpass filter 0.016–300 Hz with a sampling rate of 1000 Hz (EEG 1250; Nihon Koden, Tokyo, Japan). A grid electrode (4 × 5 contacts with 10 mm spacing; Unique Medical, Tokyo, Japan) was placed on the brain surface. The reference was placed on the epicranial aponeurosis. Interictal epileptiform discharge was defined as a clearly transient sharp activity with a duration of 20–200 ms and an amplitude higher than 200 μV [7], which was identified by visual inspection. The monitoring time was from 5 to 10 min. Interictal epileptiform discharges were observed not only adjacent to the frontal encephalocele but also the lateral cortex of the right frontal lobe (Figure 2A). Postoperative ECoG showed no obvious interictal epileptiform discharges at the surrounding frontal lobe (Figure 2B). The postoperative course was uneventful. Postoperative MRI showed adequate frontal lobectomy (Figure 2C). Pathological findings showed inflammatory changes in the brain tissue, suggesting an infectious brain injury. Her seizures improved after frontal lobectomy, indicating that the epileptogenic foci were not only the frontal encephalocele but also the surrounding frontal lobe.

## 3. Discussion

Seizures and intractable refractory focal epilepsy are caused by encephaloceles (or protrusions of neural contents through a bone defect or meninges). Ramos et al. reported that of 267 patients with epilepsy associated with encephaloceles, 262 had encephaloceles located in the temporal lobe (98.13%), three in the frontal lobe (1.12%), one in the parietal lobe (0.37%), and one in the occipital lobe (0.37%) [3]. Previous reports have shown that epileptogenic encephaloceles are most frequently located in the temporal lobe and rarely occur in the frontal lobe [3,5]. Encephaloceles are increasingly recognized as a potential cause of medically refractory epilepsy; surgical treatment with removal of the encephalocele and repair of the defect are needed [6].

We reviewed and summarized previously reported cases of epilepsy due to frontal lobe encephalocele [5,8,9,10,11,12,13,14] (Table 1). Eight cases of frontal lobe encephalocele have been reported in English literature, of which three were reported as acquired. The presentation was typically in the middle age (mean age 40.8 years), and all cases were female. Five patients underwent encephalocele resection, and the postoperative course was seizure-free in all the surgical cases. However, our case was different from previous cases in that the seizures did not improve despite encephalocele resection. Intraoperative ECoG in the second surgery showed an extensive focus of epilepsy, which suggests that there were epileptogenic foci not only in the encephalocele, but also in the surrounding brain region. Therefore, the initial surgery for encephalocele resection did not cure her epilepsy. Regarding the cause of the extensive foci, it is possible that meningitis or local infection around the encephalocele occurred due to CSF rhinorrhea. Bacterial meningitis develops in 19% of CSF rhinorrhea cases [15]. Meningitis often causes epilepsy, and epilepsy following meningitis is frequently refractory [10]. In general, a history of infectious brain insult is related to poor seizure outcomes, which indicated that infectious events can cause the epileptogenic zone to extend to the neighboring or remote region [16,17].

Pathological findings of the encephalocele and surrounding brain showed inflammatory changes, although a blood examination did not show an obvious increase in the inflammatory response, including C-reactive protein level and white blood cell count. The patient had a frontal encephalocele with a transient history of CSF rhinorrhea, which might lead to CNS infection, especially in the encephalocele and surrounding frontal lobe. Our hypothesis is that the patient may have had a wider epileptogenic focus due to the spread of inflammation caused by the encephalocele followed by CSF rhinorrhea; however, CNS infection seemed to be focal and self-resolving due to no persistent CSF leakage.

In previous studies, intraoperative ECoG has been reported as a useful tool for identifying epileptogenic foci during surgery [18]. Additionally, De Souza et al. demonstrated the usefulness of stereoencephalography (SEEG) methodology in minimizing the volume of temporal lobe resection for epilepsy due to temporal lobe encephalocele without compromising seizure and neuropsychological outcomes [19]. Therefore, recent advances in stereotactic depth electrode implantation may provide a better option for intracranial EEG monitoring in future cases, which would allow us to identify the epileptogenic zone more precisely for better outcomes. Although all previous reports have focused on temporal lobe encephaloceles, frontal lobe encephaloceles may also have better identification of epileptic zones and better surgical outcomes owing to the use of their modality. If the epileptogenic focus is suspected to be wider than the encephalocele, intracranial ECoG (intraoperative or extraoperative) may be useful for encephalocele-related epilepsy. It is crucial to confirm not only the symptoms of seizures, neuroimaging, and EEG, but also the history of CNS infection, which can widen the epileptogenic foci for encephalocele-related epilepsy.

In conclusion, we presented a case of refractory epilepsy due to frontal lobe encephalocele, which required additional surgery for surrounding frontal lobectomy for adequate seizure control. To our knowledge, this is the first case of encephalocele plus epilepsy in which seizures were not cured with frontal encephalocele resection alone. We speculated that the encephalocele and surrounding brain inflammation following CSF rhinorrhea could expand the focus of epilepsy. Encephalocele plus epilepsy should be considered when surgical treatment of epilepsy related to encephalocele is planned. Furthermore, various modalities, including intraoperative and extraoperative ECoG, may contribute to successful treatment.

## Figures and Tables

**Figure 1 brainsci-13-00115-f001:**
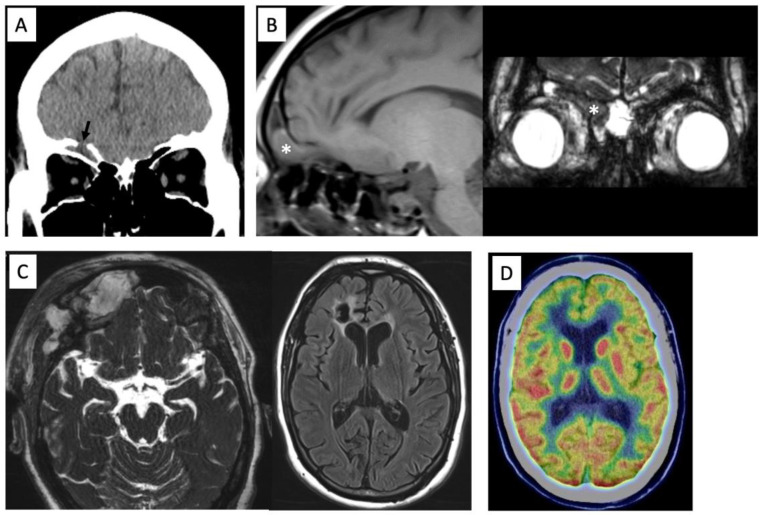
Preoperative computed tomography showing the anterior skull base defect (arrow) (**A**). MR images showing a prolapsed right rectal gyrus into the frontal sinus on T1-weighted image and constructive interference in steady state (CISS) (*) (**B**). Postoperative MRI showing adequate removal of the encephalocele and scar in the right frontal lobe on CISS and FLAIR images (**C**). 18F-fluorodeoxyglucoce PET showing hypometabolism in right lateral frontal cortex (**D**).

**Figure 2 brainsci-13-00115-f002:**
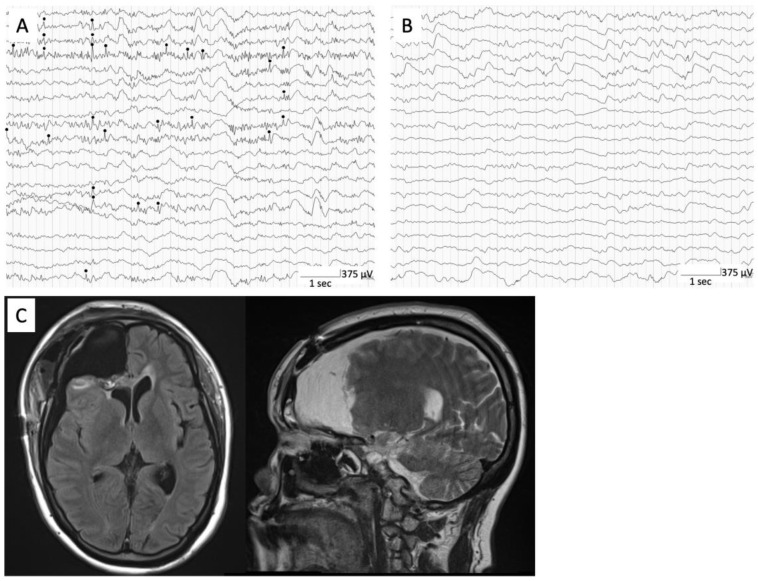
Intraoperative ECoG showing interictal epileptiform discharges (black circle) at right frontal lateral cortex (**A**). Postoperative intraoperative ECoG showing no obvious interictal epileptiform discharges at surrounding frontal lobe (**B**). Postoperative MRI showing adequate frontal lobectomy (**C**).

**Table 1 brainsci-13-00115-t001:** Reported cases of epilepsy due to frontal encephalocele.

Author & Year	Age/Sex	Location	Congenital/Acquired	Treatment	Outcome
Scully et al., 1989	63/female	Left	Congenital	Encephalocele resection	Seizure free
Guettat et al., 1998	32/female	Right	Acquired	n/a	n/a
Eichler et al., 2005	55/female	Right	Acquired	Medication	Seizure free
Mandl et al., 2007	43/female	Bilateral	Acquired	Encephalocele resection	Seizure free
Morley et al., 2008	48/female	Right	n/a	n/a	n/a
Faulkner et al., 2010	32/female	Right	n/a	Encephalocele resection	Seizure free
Ammar et al., 2012	38/female	Right	n/a	Encephalocele resection	Seizure free
Busic et al., 2015	57/female	Right	n/a	Encephalocele resection	Seizure free
Present case	44/female	Right	Acquired	Encephalocele resection Frontal lobectomy	Unchanged Improved

## Data Availability

Not applicable.

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
