# Peer review of "Frontal Encephalocele Plus Epilepsy: A Case Report and Review of the Literature"

_brainsci, 2023, doi:10.3390/brainsci13010115_

Round 1

Reviewer 1 Report

I am a neurologist with a subspecialisation in intensive care unit and I work in a hospital-teaching division of epileptology.

Yamazaki et al provides data about epilepsy related to frontal encephalocele. They discuss the case of a 44-year-old woman with an epilepsy associated with a frontal encephalocele. The epilepsy remains difficult to treat despite surgical resection of frontal encephalocele.

The authors should be congratulated for having provided these interesting data on this clinical problem. Thank you for the opportunity to comment this work. The reading is easy.

Some parts of the manuscript must be improving (major points):

1. I have some concerns about the authors' hypothesis of epileptogenic focus. They hypothesize that "meningitis or local infection around the encephalocele" could be the original event of epileptogenesis. So, how the authors explain the spontaneous evolution of this CNS "local infection" in absence of antimicrobial treatment?

2. It is not acceptable to present EEG or EcoG epochs without a precise definition of their characteristics : duration of epochs? filters? amplitude? derivations?

In the actual version, the readers (including me) could not understand the figure 2.

Minor points:

3. You may precise in figure 2, panel A, with an arrow or a circle, the epileptiform activities, especially for non-specialist readers.

4. What do you mean with "Literature of review" in the title ? ...

Ethical: No comment.

Statistics: NA.

Author Response

We would like to thank the reviewer for giving us your comments on this manuscript. The following is a point-by-point response to the reviewer’s comments and suggestions.

I am a neurologist with a subspecialisation in intensive care unit and I work in a hospital-teaching division of epileptology.

Yamazaki et al provides data about epilepsy related to frontal encephalocele. They discuss the case of a 44-year-old woman with an epilepsy associated with a frontal encephalocele. The epilepsy remains difficult to treat despite surgical resection of frontal encephalocele.

The authors should be congratulated for having provided these interesting data on this clinical problem. Thank you for the opportunity to comment this work. The reading is easy.

Response: We appreciate the reviewer’s positive comment.

Some parts of the manuscript must be improving (major points):

  1. I have some concerns about the authors' hypothesis of epileptogenic focus. They hypothesize that "meningitis or local infection around the encephalocele" could be the original event of epileptogenesis. So, how the authors explain the spontaneous evolution of this CNS "local infection" in absence of antimicrobial treatment?

Response: We appreciate the reviewer’s constructive comment and concern. The patient had a transient history of CSF rhinorrhea, but no obvious clinical symptoms of meningitis. We presumed that CNS infection seemed to be focal and self-resolved because CSF leakage had not persisted. We added the following sentences.

Page 2, line 48-49: there were no obvious clinical symptoms suggesting meningitis.

Page 4, line 127-128: however, CNS infection seemed to be focal and self-resolving due to no persistent CSF leakage.

  1. It is not acceptable to present EEG or EcoG epochs without a precise definition of their characteristics : duration of epochs? filters? amplitude? derivations?

In the actual version, the readers (including me) could not understand the figure 2.

Response: I apologize for our mistakes. As the reviewer commented, we added the sentences and reference about method of ECoG and the definition of Interictal epileptiform discharge.

Page2, line: 58-63: ECoG was recorded with a bandpass filter 0.016–300 Hz with a sampling rate of 1000 Hz (EEG 1250; Nihon Koden, Tokyo, Japan). A grid electrode (4 x 5 contacts with 10 mm spacing; Unique Medical, Tokyo, Japan) was placed on the brain surface. The reference was placed on the epicranial aponeurosis. Interictal epileptiform discharge was defined as a clearly transient sharp activity with a duration of 20–200 msec and amplitude higher than 200 μV [7], which was identified by visual inspection. The monitoring time was from 5 to 10 minutes.

Minor points:

  1. You may precise in figure 2, panel A, with an arrow or a circle, the epileptiform activities, especially for non-specialist readers.

Response: We appreciate the reviewer’s advice. I added black circle on the obvious epileptiform activities in Figure 2.

  1. What do you mean with "Literature of review" in the title ? ...

Response: We appreciate the reviewer’s comment. We reviewed and summarized the manuscripts about epilepsy due to frontal encephalocele. Therefore, I want to use “literature of review” in this title. I am happy if the reviewer accepts my opinion. 

Ethical: No comment.

Statistics: NA.

Again, we thank reviewer for your time, suggestions, and comments.

Reviewer 2 Report

Dear Authors,

Your paper is well described, however, you suggest that epilepsy is related to the encephalocele.

1. Have you thought of the idea that it could be a case of  epilepsy unrelated to the encephalocele? you did lobectomy.

2. Have you done a preoperative surgery for the epilepsy to observe the sensors' activity? 

Author Response

We would like to thank the reviewer for giving us your comments on this manuscript. The following is a point-by-point response to the reviewer’s comments and suggestions.

Dear Authors,

Your paper is well described, however, you suggest that epilepsy is related to the encephalocele.

Response: We appreciate the reviewer’s positive comment.

  1. Have you thought of the idea that it could be a case of epilepsy unrelated to the encephalocele? you did lobectomy.

Response: We appreciate the reviewer’s constructive comment and concern. We diagnosed that the patient had acquired focal epilepsy due to frontal encephalocele because of her age, clinical history, clinical symptoms such as CSF rhinorrhea, and imagings. EEG also showed the epileptogenic focus was concordant to the right frontal lobe. We think the etiology of her epilepsy can be structural due to frontal encephalocele.

  1. Have you done a preoperative surgery for the epilepsy to observe the sensors' activity? 

Response: We appreciate the reviewer’s comment. We did not perform intraoperative and extraoperative ECoG in first surgery (encephalocele resection). However, intraoperative ECoG was useful in second surgery (frontal lobectomy), we discussed that ECoG can be important when we deal with encephalocele epilepsy (Page 4, line 129-141).

Again, we thank reviewer for your time, suggestions, and comments.

Round 2

Reviewer 1 Report

Dear Authors, 

I appreciate the modifications made concerning ECoG recording. However, I always have some concerns about the hypothesis of local and self-resolved infection of CNS. Please precise that it was "our hypothesis" in discussion section (line 106).

I still do not understand the title "litterature of review". I suppose you mean "review of the litterature", for example ? I do not find this term in Pubmed in a brief research ...

Author Response

We would like to thank the reviewer for giving us your comments on this manuscript. The following is a point-by-point response to the reviewer’s comments and suggestions.

I appreciate the modifications made concerning ECoG recording. However, I always have some concerns about the hypothesis of local and self-resolved infection of CNS. Please precise that it was "our hypothesis" in discussion section (line 106).

Response: We appreciate the reviewer’s constructive comment and concern. As the reviewer said, this sentence is our hypothesis. We added “our hypothesis” in discussion section.

Page4, line: 126-129: Our hypothesis is that the patient may have had a wider epileptogenic focus due to the spread of inflammation caused by the encephalocele followed by CSF rhinorrhea; however, CNS infection seemed to be focal and self-resolving due to no persistent CSF leakage.

I still do not understand the title "litterature of review". I suppose you mean "review of the litterature", for example ? I do not find this term in Pubmed in a brief research ...

Response: I apologize for our mistakes very much despite the English editing services revising this manuscript. As the reviewer commented, wrong: literature of reviewcorrect: review of the literature.

I changed the title: Frontal encephalocele plus epilepsy: a case report and review of the literature. I appreciate the reviewer for giving us your comment to improve our manuscript.

Again, we thank the reviewer for your time, suggestions, and comments.

Reviewer 2 Report

Thank you for your revision. 

Author Response

We would like to thank the reviewers for giving us your comments on this manuscript.

We appreciate the reviewer’s positive comments, suggestions, and your time on our manuscript.